# Distinct temporal diversity profiles for nitrogen cycling genes in a hyporheic microbiome

William C. Nelson[1]*, Emily B. Graham[1], Alex R. Crump[2], Sarah J. Fansler[1], Evan V. Arntzen[1], David W. Kennedy[1], James C. Stegen[1]

**1** Pacific Northwest National Laboratory, Richland, Washington, United States of America, **2** Department of Soil and Water Systems, University of Idaho, Moscow, Idaho, United States of America

* william.nelson@pnnl.gov

## Abstract

Biodiversity is thought to prevent decline in community function in response to changing environmental conditions through replacement of organisms with similar functional capacity but different optimal growth characteristics. We examined how this concept translates to the within-gene level by exploring seasonal dynamics of within-gene diversity for genes involved in nitrogen cycling in hyporheic zone communities. Nitrification genes displayed low richness—defined as the number of unique within-gene phylotypes—across seasons. Conversely, denitrification genes varied in both richness and the degree to which phylotypes were recruited or lost. These results demonstrate that there is not a universal mechanism for maintaining community functional potential for nitrogen cycling activities, even across seasonal environmental shifts to which communities would be expected to be well adapted. As such, extreme environmental changes could have very different effects on the stability of the different nitrogen cycle activities. These outcomes suggest a need to modify existing conceptual models that link biodiversity to microbiome function to incorporate within-gene diversity. Specifically, we suggest an expanded conceptualization that 1) recognizes component steps (genes) with low diversity as potential bottlenecks influencing pathway-level function, and 2) includes variation in both the number of entities (*e.g.* species, phylotypes) that can contribute to a given process and the turnover of those entities in response to shifting conditions. Building these concepts into process-based ecosystem models represents an exciting opportunity to connect within-gene-scale ecological dynamics to ecosystem-scale services.

## Introduction

High microbial diversity has been observed in almost all environments that have been examined [1]. It is widely believed that this diversity provides functional stability to ecosystems experiencing fluctuations in environmental conditions by the presence of organisms having overlapping functional capabilities but different conditions under which they optimally

**Funding:** This research and all authors were supported by the US Department of Energy, Office of Biological and environmental Research (https://www.energy.gov/science/ber/biological-and-environmental-research), as part of Subsurface Biogeochemical Research Program's (https://www.doesbr.org) Scientific Focus Area (SFA) at the Pacific Northwest National Laboratory (PNNL). PNNL is operated for DOE by Battelle under contract DE-AC06-76RLO 1830. Sequencing was done at the DOE Joint Genome Institute under Community Science Project 1781 awarded to J.C.S. The funders had no role in study design, data collection and analysis, decision to publish, or preparation of the manuscript.

**Competing interests:** The authors have declared that no competing interests exist.

function [2–8]. In a fluctuating environment, conditions that impair the growth of some populations will stimulate the growth of others, and overall community function is maintained. Maintenance of higher diversity therefore allows a community to respond more rapidly to a disturbance or environmental shift and reduces its dependence on (or susceptibility to) recruitment of new organisms to fill vacant niches. The dynamics of diversity at the functional gene level, however, have not been well explored.

Cooperative metabolism in natural microbial communities has long been suspected, but only recently have metagenomic studies revealed its extent. The component steps (*i.e.*, individual enzyme-catalyzed reactions) of complex metabolic pathways, such as denitrification, sulfur oxidation, and organic carbon degradation, have been observed to be distributed across multiple organisms more frequently than they are co-resident in a single organism [9, 10]. Distributed metabolism likely reflects efficiency gains from specialization and division of labor [11]. This partitioning, however, puts component steps of critical ecosystem processes under different selective pressures, according to which organism encodes them. Temporal dynamics of diversity and abundance may, therefore, vary significantly across component steps.

Nitrogen cycling is an excellent and ubiquitous example of a complex, distributed process. A generalized model of the dominant N processes in the HZ (Fig 1) includes conversion of $NH_4^+$ to $NO_3^-$ (i.e., *nitrification*) in oxic regions, which is coupled to carbon (C) fixation, and reduction of $NO_3^-$ coupled to the oxidation of organic carbon (OC) in anoxic regions. This latter process of *denitrification* yields $N_2$ gas, removing N from the system. Nitrite ($NO_2^-$) is an intermediate common to both processes. While complete denitrifier organisms, such as *Pseudomonas aeruginosa* and *Parcoccus denitrificans*, have been isolated and described, it has long been suspected that many organisms encode partial pathways and can act in concert to cycle nitrogen between its reduced and oxidized forms [12]. More recently, genome sequence data from both isolates and environmental samples has shown that many organisms encode various subsets of denitrification activities [9, 13]. Several previous studies have investigated the abundance and distribution of nitrogen cycling activities in environmental microbiomes [14–19], none yet have specifically tracked the diversity of individual gene families that comprise nitrogen transformation pathways across fluctuating environmental conditions.

Here we take advantage of seasonal shifts in hydrology and aqueous geochemistry within a hyporheic zone system that have been shown to alter microbial community structure [20, 21], and examine the temporal dynamics of diversity within major N-cycling genes encoding steps in nitrification and denitrification. Some component steps consistently showed very low diversity, while others displayed significant temporal variation in the level of diversity and turnover in the contributing phylotypes across divergent environmental conditions. The observed heterogeneity through time and across component steps indicates that predictive ecosystem models that explicitly represent microbial communities should account for variation in and dynamics of within-gene diversity of component steps of key processes.

## Results

### Seasonal environmental changes

Sediment communities from the hyporheic zone of the Columbia River along the Hanford Reach were sampled from April 30, 2014 to November 25, 2014, using sand packs deployed at three equivalent hyporheic zone locations approximately 100m apart along the river (T2, T3, and T4) for six weeks at a time [22]. Water chemistry data taken in parallel at the three sites showed similar, yet not identical temporal patterns. A mid-year shift in hydraulic regime was observed, with higher influx of surface water in the spring resulting in higher levels of dissolved organic carbon (measured as non-purgeable organic carbon) (NPOC) (0.8–1.0 mg/L)

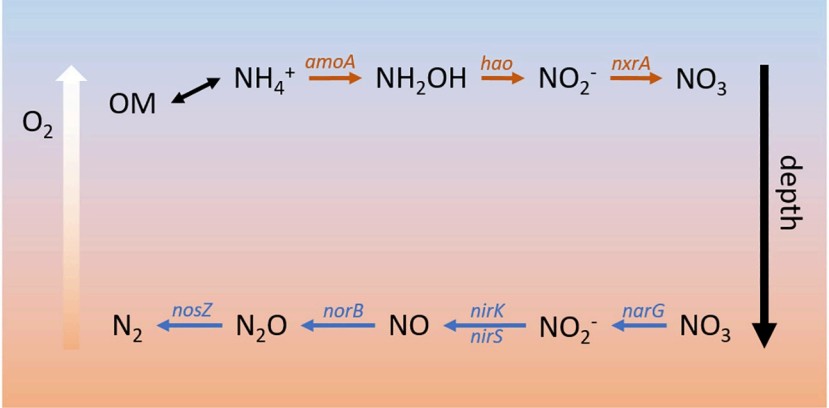

**Fig 1. Major nitrogen transformations in the hyporheic zone (HZ).** Upper layers (closer to the surface channel) of the hyporheic zone contain more oxygen ($O_2$) and organic matter (OM). Under these conditions, *nitrification* (orange arrows) occurs. Ammonium ($NH_4^+$) is released from by OM breakdown and converted to nitrate ($NO_3$) through hydroxylamine and nitrite ($NO_2^-$) intermediates. This process has been linked with carbon fixation, increasing organic carbon (OC). Aerobic respiration depletes $O_2$, causing deeper regions of the HZ to become hypoxic or anaerobic. Under these conditions, *denitrification* (blue arrows) converts nitrate to nitrogen gas ($N_2$) through nitrite, nitric oxide (NO) and nitrous oxide intermediates, and provides an electron acceptor for catabolism of OM.

(Fig 2A) and low levels of nitrate (10–15 µM) (Fig 2B), transitioning to a more groundwater-influenced condition in the fall, increasing the nitrate concentrations (up to 300 µM) and decreasing NPOC concentration (down to <0.4 mg/L). Because the groundwater in this system is oxic, the DO concentration was fairly constant for the duration of sampling, ranging from ~60–100% saturation (Fig 2C). The water temperature followed expected seasonal trends, warming in the summer and cooling in the fall (Fig 2D). Sampling times were categorized as early (Apr 30 through Jul 22) or late (Sep 2 through Nov 30), based on these observations.

## Organism-level diversity

Organismal diversity was measured by 16S rRNA V4 amplicon sequence analysis and extraction and assembly of *rplB* gene sequences from the metagenomic data sets (Fig 3). As reported previously [20], species richness correlated best with water temperature. Diversity, as measured by the inverse Simpson statistic, was high and mirrored species richness, suggesting high evenness. Two late samples, October 14 and November 25, showed high richness but low diversity, driven by a bloom of Bacteroidetes species.

## Diversity of N-cycling genes

The temporal phylogenetic profile of each gene of interest was examined to elucidate the richness and diversity of genes comprising the nitrification and denitrification processes. Metagenomic reads containing sequence from the genes of interest were extracted from the total data set and assembled to yield partial and full-length gene sequences (Supplementary Data 1). Phylogeny was determined for each assembled sequence, and phylotypes were defined at 90% amino acid sequence identity, since that level of similarity is typical between organisms of the same genus [23]. Richness was quantified for each gene as the number of distinct phylotypes identified. It was expected that detectable gene diversity would be considerably lower than organismal diversity, since 1) these activities are encoded by a subset of organisms, and 2) the assembly protocol is less sensitive than amplicon analysis, and thus only genes from abundant organisms are likely to be detected. The relative abundance of each phylotype was estimated

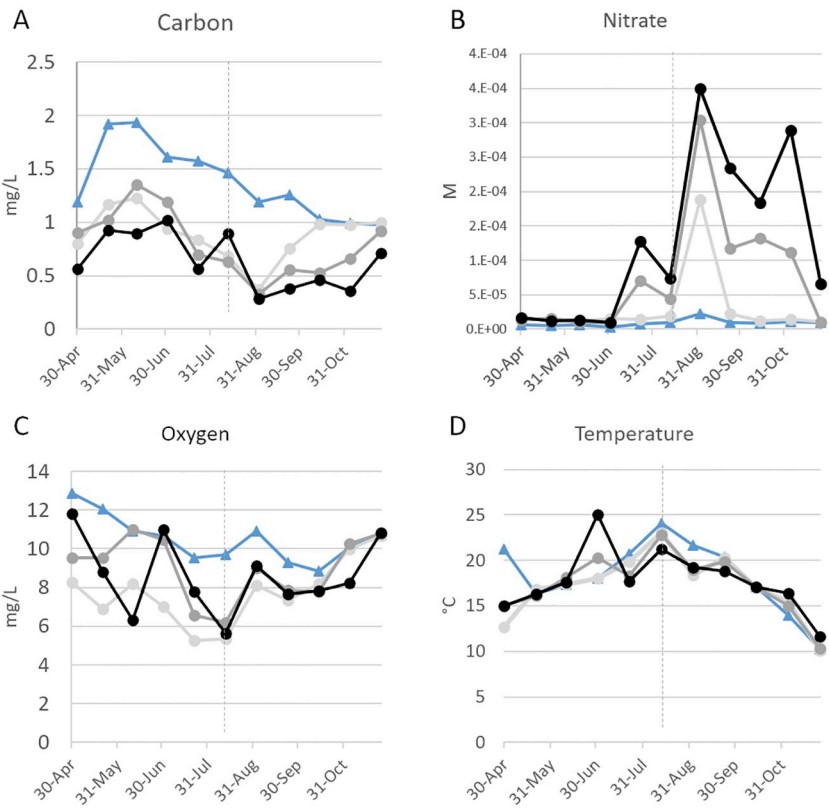

**Fig 2. Water chemistry and temperature of sampled sites.** Piezometer T2, light gray; piezometer T3, dark gray; piezometer T4, black. For comparison, data for adjacent river water is presented (blue). The vertical dotted line indicates the date at which the hyporeheic zone hydraulic regime changes from surface water intrusion to groundwater discharge.

from the summed assembly coverage of the member genes. Temporal diversity dynamics (turnover) were assessed by calculating the mean variance of relative abundance for phylotypes across time.

Distinct diversity and turnover patterns were observed for each gene. The *narG* and *nosZ* genes (Figs 4 and 5, summarized in Table 1), encoding the first and last steps of the denitrification process, respectively, had higher phylotype richness than the other nitrogen cycle genes examined (for nosZ vs norB, Welch's t-test p-value = 0.0014, df = 13.587), and their phylotype profiles had equivalent stability (Levene test p-value = 0.1277). While the *nirK/nirS* (distinct types of nitrite reductase) (Fig 6) and *norB* family (nitric oxide reductase) (Fig 7) had lower richness, their phylotype profile variability was significantly higher than for *narG* (Levene test p-values = 0.00003, 0.0001, respectively), and were near significance for *nosZ* (Levene test p-values = 0.0113, 0.0609 for *nosZ*I and *nosZ*II). Both genes encoding activities involved in nitrification had extremely low phylotype diversity, *amoA* (ammonia oxidase) with 2 phylotypes, one bacterial and one archaeal, and *nxrA* (nitrite oxidase alpha subunit) having 7 observed, but one overwhelmingly dominant phylotype (Fig 8A). The low richness for *amoA* (Fig 8B) exaggerates the phylotype abundance variance values, thus we consider the low richness to be the significant aspect of the *amoA* gene.

Nitrogen gene diversity was largely dependent upon a temporally consistent pool of taxa. An examination of cumulative phylotype richness (Fig 9A) showed an increase in the number phylotypes detected for almost all the target gene families in the spring, with limited increase

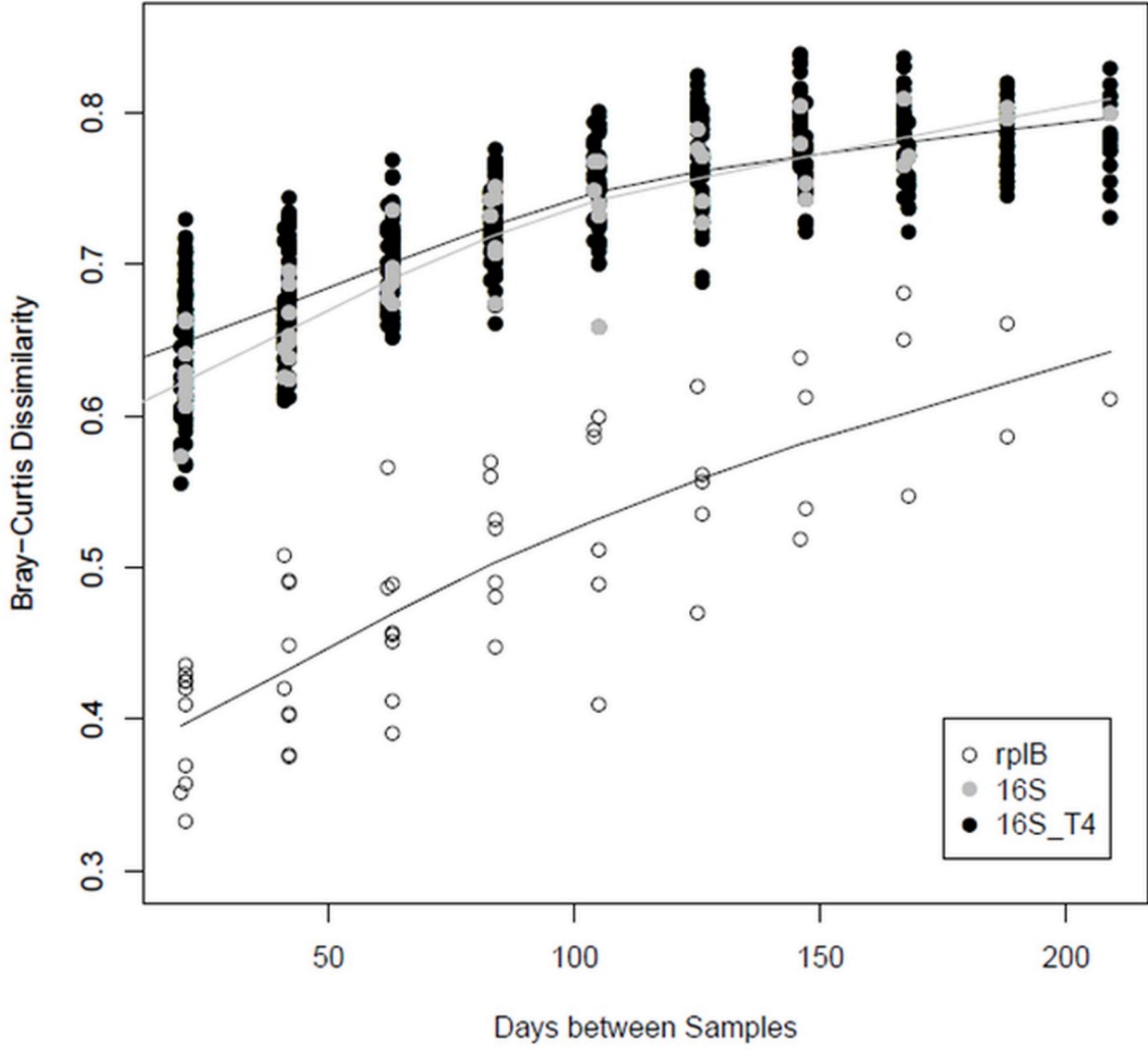

**Fig 3. Sediment microbial community continuously changes across the year.** Distance-decay plot of all 16S rRNA amplicon data ("16S"), amplicon data from only site T4 ("16S_T4"), and *rplB* genes ("rplB") assembled from all metagenomes.

thereafter. Importantly, cumulative richness curves generated using unique sequences, rather than phylotypes, have equivalent shape (data not shown). An analysis of cumulative diversity (inverse Simpson) showed that the increase in the number of phylotypes had a proportional effect on diversity except in the cases of *narG*, where the large increase in phylotypes only translates to a modest increase in diversity, and *nxrA* where the small increase in richness had no effect on diversity (Fig 9B). This was due to the additional phylotypes having low relative abundance. The *nirKS* family showed an initial decrease in diversity despite an increase in the number of phylotypes, and a subsequent increase in diversity with no further increase in richness. This increase in diversity is due to increasing evenness amongst the various phylotypes present. The *nosZ* family was the only one to demonstrate consistent increases in diversity due to introduction of new phylotypes.

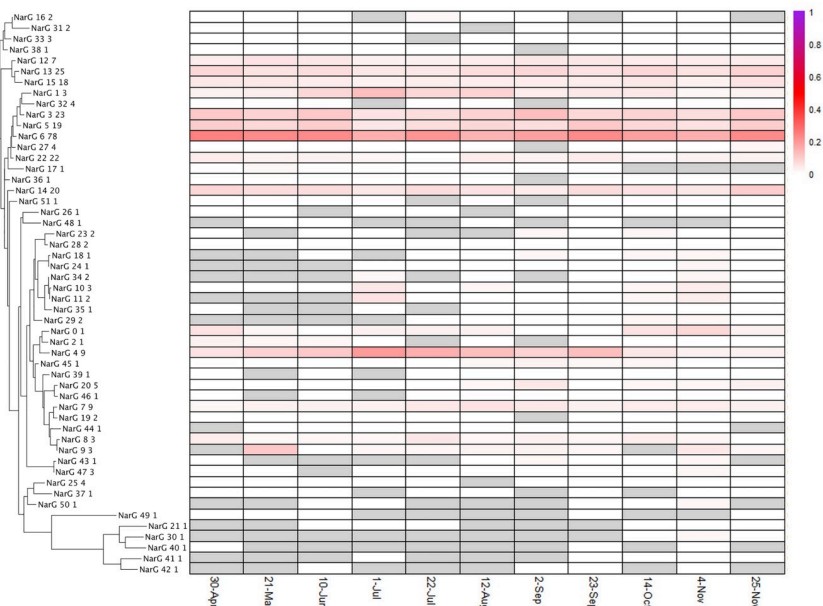

**Fig 4. NarG phylotype distributions.** Heatmap indicates the relative contribution of each phylotype (clustered at 90% AAID) to the total count; phylogenetic tree to the left of the heatmap demonstrates the diversity of phylotypes present. Gray shading indicates no observation of the phylotype at that timepoint.

## Abundance of N-cycling genes

To assess temporal changes in the overall abundances of genes involved in denitrification and nitrification, the sets of all (*i.e.*, unassembled) metagenomic reads containing sequence from the genes of interest were enumerated, and the representation of each gene within the

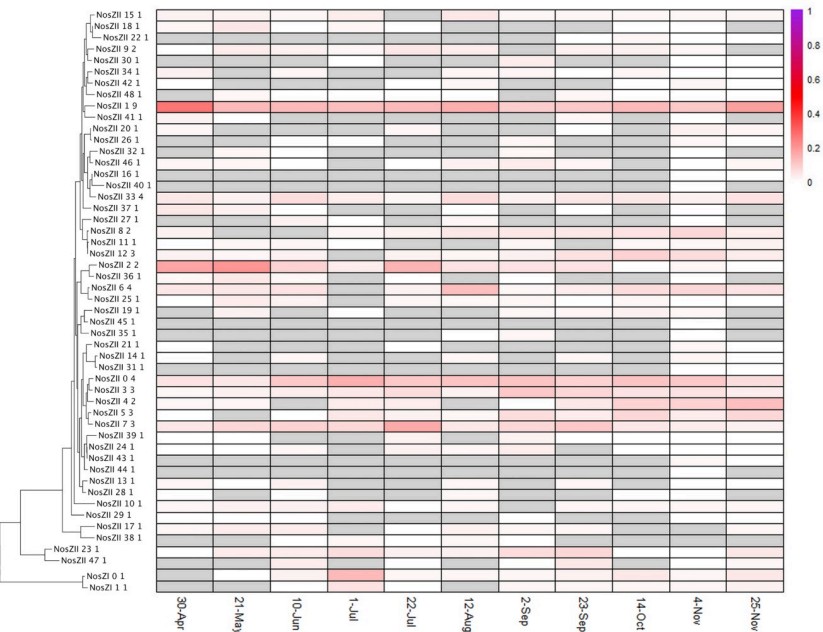

**Fig 5. NosZI and NosZII phylotype distributions.** See Fig 4 for description of display.

**Table 1. Observed richness and abundance variance.**

| Gene family | Phylotype richness | Mean abundance variance |
| --- | --- | --- |
| narG | 52 | 0.000183 |
| norB | 31 | 0.000624 |
| nirKS | 23 | 0.001095 |
| nosZ | 51 | 0.000395 |
| nxrA | 7 | 0.000047 |
| amoA | 2 | 0.089783 |
| rplB | 124 | 0.000112 |

community was normalized across samples using counts of the conserved, single-copy *rplB* gene as a proxy for number of individuals sampled. Although gene abundances were relatively constant over time, the average abundances differed widely between genes. The *narG* gene, the first step in denitrification, was observed to be in 25–30% of the population, while the *nirK/ nirS* was represented in 35–45% of the population, and *norB* in 14–18% (Fig 10). Nitrous oxide reductase genes were present in ~25% of the populations, however it is of note that the dominant form was *nosZ*II (also referred to in the literature as the 'atypical *nosZ*'), a distinct family of nitrous oxide reductases typically found in non-denitrifying organisms [13, 24, 25]. Nitrification genes showed more of a seasonal shift in abundance. The *amoA* gene, summing both

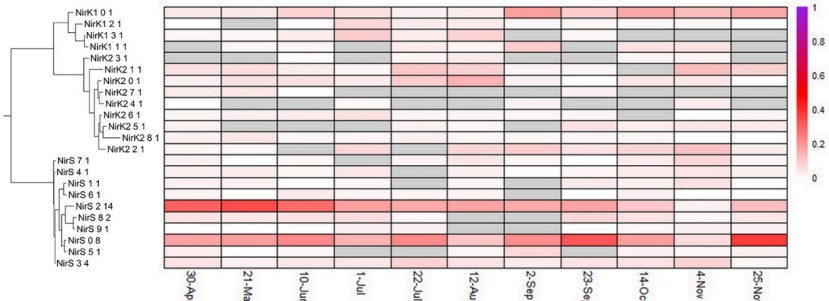

**Fig 6. NirK and NirS phylotype distributions.** See Fig 4 for description of display.

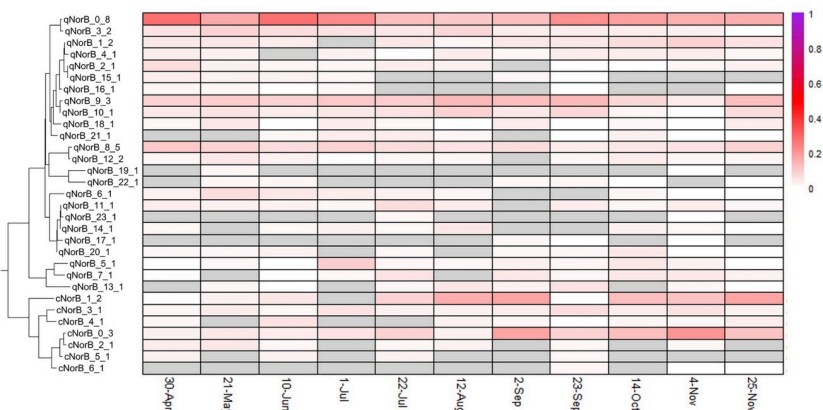

**Fig 7. NorB phylotype distributions.** See Fig 4 for description of display.

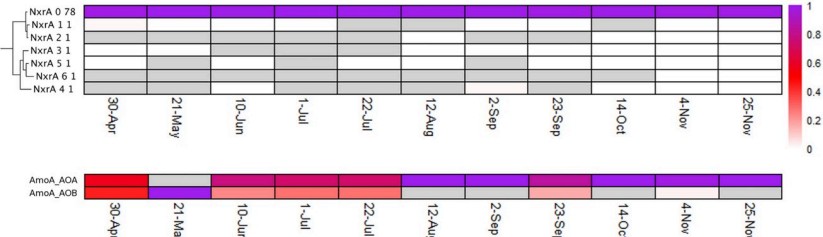

**Fig 8. A) NxrA and B) AmoA phylotype distributions.** See Fig 4 for description of display.

the bacterial and archaeal versions, showed a low constant abundance of ~5% in early time points, and increased up near 30% late in the year. Unexpectedly, *nxrA* showed little correlation with *amoA*, displaying a trend of gradual increase, ranging from 5% to 18%, early, and constancy late.

## Environmental drivers

Regression analysis was performed to determine which, if any, of the environmental parameters measured was associated with changes in diversity for the genes of interest. Water temperature, dissolved oxygen (DO), dissolved organic carbon (measured as non-purgeable organic carbon, NPOC), and chloride (Cl⁻) measurements were used. Cl⁻ is a conservative indicator of the ratio of surface- to groundwater content in the hyporheic zone of the study system [26]. Other measured constituents, $NO_3^-$ and $SO_4^-$ had strong positive correlations with Cl⁻ (S1 Fig). Correlations between diversity (inverse Simpson), richness, and abundance were tested

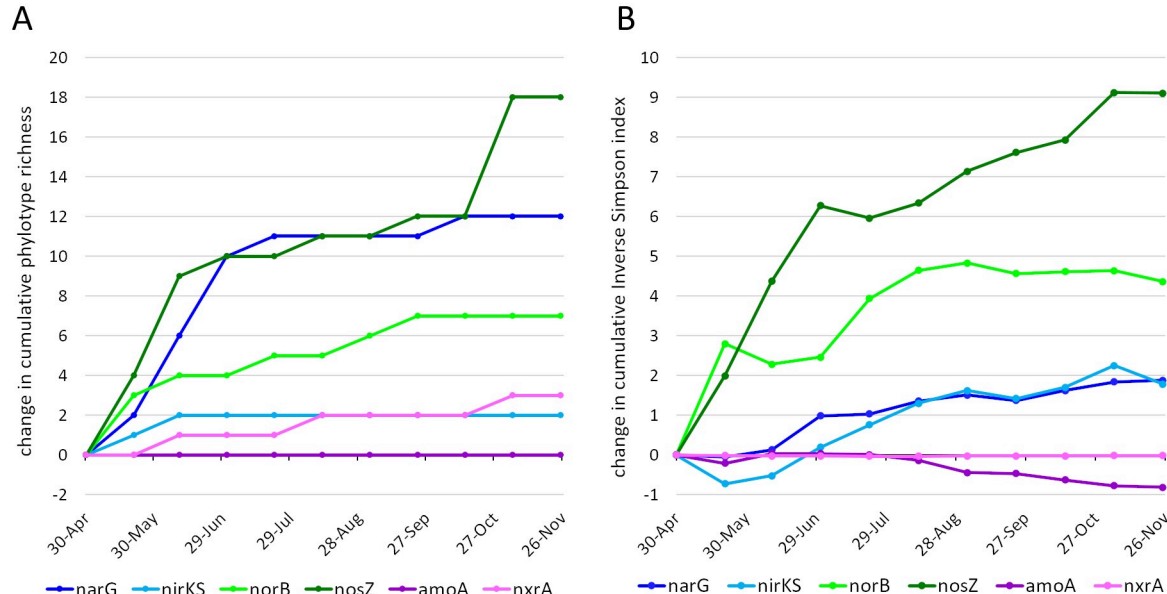

**Fig 9. Cumulative diversity measures over time.** Phylotype richness (A) and the inverse Simpson statistic (B) were calculated cumulatively (*i.e.*, combining data from each time point with all previous timepoints) for each gene or functional gene class (*nirK* and *nirS* counts were combined; archaeal and bacterial *amoA* types were combined). The data is presented as the difference from the initial (April 30) state. Most genes' richness values plateau, indicating sample-to-sample changes in diversity are within a finite pool of phylotypes. Diversity increase indicates either introduction of new phylotypes or increases in evenness across existing phylotypes. The decreases in diversity observed for *amoA* and *nirKS* are driven by changes in relative abundance resulting in a decrease in evenness.

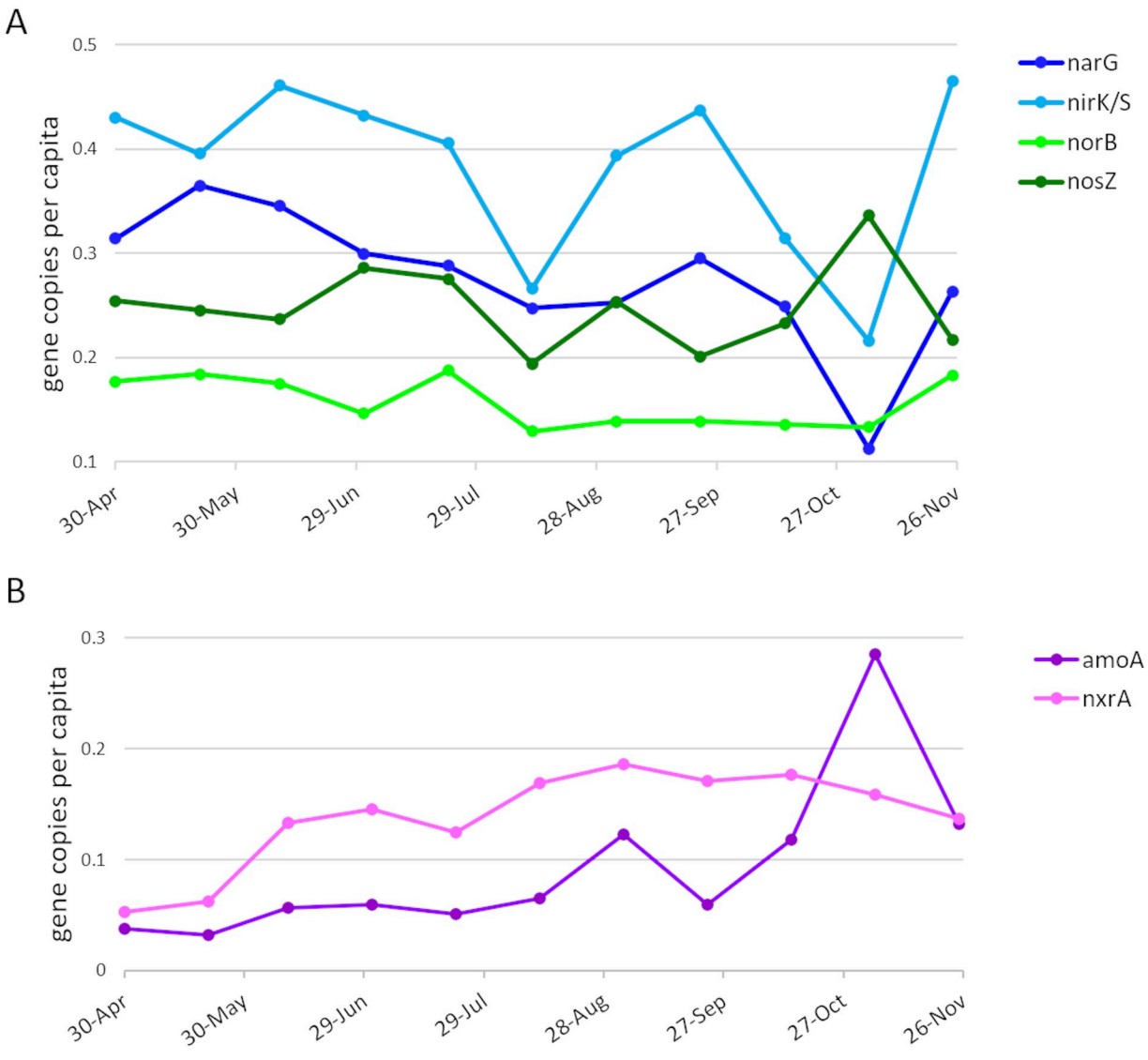

**Fig 10. Per-capita abundance of denitrification and nitrification genes.** Reads per kilobase of gene length per million reads (RPKM) for each gene was normalized against the RPKM for the *rplB* gene as a proxy for the number of individuals sampled. (A) Denitrification genes. (B) Nitrification genes.

against the environmental parameters. The strongest relationships were with groundwater content (using Cl⁻ as a proxy), with denitrification genes *narG* ($R^2 = 0.38$; p = 0.04) and *nosZ* ($R^2 = 0.50$; p = 0.02) increasing in diversity (S2 Fig), nitrification genes *amoA* ($R^2 = 0.41$; p = 0.03) and *nxrA* ($R^2 = 0.44$; p = 0.03) increasing in abundance (S3 Fig), and *narG* ($R^2 = 0.47$; p = 0.02) decreasing in abundance. Groundwater showed weaker correspondence with increasing richness of *nxrA* ($R^2 = 0.29$; p = 0.09), decreasing richness of *nirKS* ($R^2 = 0.27$; p = 0.10) (S4 Fig), and decreasing abundance of *norB* ($R^2 = 0.35$; p = 0.06). NPOC had strongest correlations with the nitrification genes, showing a negative relationship with *nxrA* diversity ($R^2 = 0.31$; p = 0.08), and a positive relationship with *nirKS* richness ($R^2 = 0.39$; p = 0.04) and *narG* abundance ($R^2 = 0.30$; p = 0.08). Temperature had a significant negative relationship with *nxrA* diversity ($R^2 = 0.33$; p = 0.08) and richness ($R^2 = 0.45$; p = 0.03).

## Discussion

Shade *et al.*, in their review of microbial resistance and resilience, suggest that there is "no 'one-size fits all' response of microbial diversity and function to disturbance." [8]. While this perspective is undoubtedly true, it leaves open the possibility that there are general patterns or rules that govern particular subsets or components of microbial communities. Here we begin to look for such patterns at a deeper level than previously examined by exploring dynamics in gene abundance and diversity within important biogeochemical processes in response to seasonal environmental changes. Building from recent work showing that component steps in biogeochemical processes are encoded by separate microbial taxa [9, 10], we hypothesized that within-gene diversity varies between component steps, and further that temporal dynamics of diversity would vary between steps. Our metagenomic data from a dynamic groundwater-surface water mixing zone were consistent with this hypothesis and demonstrated that within-gene diversity, and the dynamics of that diversity, are variable across genes. This outcome suggests that a community's taxonomic diversity or the abundance or diversity of any single (proxy) gene is not be a reliable predictor of stability in functional potential for multi-step biogeochemical processes, and that portions of the community that encode component steps with low within-gene diversity may be the most critical when considering potential decreases in function. Therefore, there is a need to shift the focus of analyses from taxonomic diversity or 'representative' gene abundances to a comprehensive understanding of within-gene diversity and dynamics across processes. Below we place these discoveries in context of previous work and point toward how they can be used to improve predictive models of system function.

### Diversity dynamics of nitrification genes

The nitrification process showed low diversity for both steps examined, leading to the possibility that these activities are susceptible to loss or diminished function. Nitrification was originally described as a cooperative process, requiring an ammonia oxidizing organism that produces nitrite and a nitrite oxidizing organism that converts the nitrite to nitrate [27]. Recently, organisms have been identified that have both activities (comammox) [28]. The range of organisms known to encode nitrification activities is narrow, although it does include both Bacteria (Nitrosomonas and Nitrospira) and Archaea (Thaumarchchaeota). The observed abundance of nitrifying organisms in sediment communities, both freshwater and marine, suggests nitrification is an important activity in the subsurface environment [18, 29, 30]. The limited taxonomic distribution of nitrification activities in the hyporheic community was expected, however the low diversity, one phylotype for *nxrA*, and one sequence apiece for the bacterial and archaeal *amoA*s is extreme. This lack of diversity suggests these activities could be unstable, given observations demonstrating that community-level functional stability increases with diversity [7, 31, 32]. However, we observed very stable abundance of these organisms across the seasonal shift in water chemistry, suggesting that the organisms encoding these activities are well adapted to the range of environmental conditions historically experienced by this community. Any extraordinary shift in biotic (*e.g.*, viruses, predation) or abiotic (*e.g.*, redox potential, temperature) conditions that selects against the small number of taxa involved in nitrification, however, could quickly degrade the community's nitrification potential. With no other apparent organisms available to supplement or take over this role, this fundamental service could be degraded or lost from this community with unknown repercussions for the microbial community and the larger ecosystem [33, 34]. Recently, nitrifiers, and in particular Archaeal nitrifiers, have been shown to be active in carbon fixation in freshwater benthic sediments [35]. Thus, loss of nitrifiers could impact coupled carbon-nitrogen cycling in the subsurface and associated river corridors.

## Diversity dynamics of denitrification genes

Denitrification genes have been identified in a broad range of taxa [36], and as such, our expectation was that within the hyporheic zone community there would be a high diversity across all component steps [37, 38]. While we did observe considerable overall abundance of all genes, the levels of richness for the genes representing the individual activities varied, ranging from 52 phylotypes for nitrate reduction (*narG*) to 23 phylotypes for nitrite reductase (*nirK* and *nirS*). This observation supports the concept that denitrification genes are distributed among members of the community as partial pathways or individual genes [14, 15]. Further, there was a surprising distribution of nitrous oxide reductase genes, with the type II form (*nos*ZII), which is typically found in non-denitrifying organisms [25], having much greater abundance and richness (49 phylotypes) than the type I form (*nos*ZI, 2 phylotypes).

Temporal variance of within-gene diversity for genes involved in both nitrification and denitrification demonstrates that the organisms encoding these activities are sensitive to different ecological selection pressures and thus different strategies are required to maintain functional potential in response to perturbation. For genes with high phylotype richness, high temporal abundance variance indicates a changing phylotype profile (*nirKS*, *norB*). These functions may be maintained through resilient microbial taxa that recover rapidly from environmental change. Conversely, low temporal variance (*narG*, *nosZ*) indicates a stable phylotype profile. These functions are maintained through resistant taxa that persist across a broad range of environmental conditions, with the possibility that the other low abundance phylotypes are capable of supplanting them should they fail under different conditions.

It is notable that while all genes associated with denitrification had high phylotype richness (in contrast to nitrification genes), the genes associated with intermediate reactions had higher temporal diversity variance than *narG* (Table 1), which encodes the initial step in denitrification (i.e., nitrate reduction). One explanation for the observed differences could be that there are different levels of competition for the substrates fueling each activity. Intermediate substrates nitrite and nitric oxide may be produced slowly and/or consumed quickly, especially considering there are multiple cellular processes for which they are intermediates and they are both toxic to cells. Supporting this contention, nitrite is typically undetectable in samples from this location, while nitrate is readily detectable [21]. Low availability would lead to high substrate competition, which could result in the increased phylotype turnover observed in *nirK*, *nirS* and *norB* genes. Modeling the redundancy provided to a process by within-gene diversity thus requires an understanding of temporal variation in the selective pressures for each gene involved.

## Influence of seasonal changes in hydrogeochemsitry

Seasonal changes in groundwater to surface water ratios appear to be a major influence on N-cycling functional potential in microbial communities. Increase in groundwater content corresponded to increasing per-capita abundance of nitrification genes and decreasing abundance/increasing diversity of denitrification genes. The *nirKS* and *norB* gene families, which displayed similar high phylotype turnover behavior, were not similar in their response to the environmental parameters measured, with *nirKS* showing a decrease in richness in response to groundwater while *norB* showed a decrease in abundance. The *narG* and *nosZ* gene families, which showed more stable profiles, both increased in diversity in response to groundwater, however, *nosZ* did so through increased richness, while *narG* likely gained evenness through reduced abundance of dominant phylotypes. Organic carbon (NPOC) had a much weaker association with gene-level metrics, relative to groundwater. A group of co-occurring organisms with a negative correlation to groundwater has been reported in this sediment system

**Fig 11. Circuit diagram of a metabolic pathway.** Steps in series convert substrates (S), to various intermediates (I1, I2), to a product (P). Redundancy is represented by parallel paths, which can be regulated individually (denoted by arrow gates). Under conditions A and B, product is produced, but by different paths, whereas under condition C, although the blue and green steps are active, neither of the orange steps are, preventing production of I2 and P.

[21]. The group is dominated by Alpha-, Beta- and Gammaproteobacteria, Bacteroidetes and Planctomycetes, the same taxa that encode nearly all of the identified denitrification genes. Strong homogenous selection was shown to be the mechanism structuring this group [20]. Taken together, these data suggest that some factor other than carbon that is within the groundwater is the selective force driving the diversity dynamics of these organisms carrying N-cycling genes. A likely candidate is the N content of groundwater, which is significantly higher than that of the surface water [26].

## Gene diversity and process resilience

Conceptualizing and studying diversity within individual gene families is a departure from the contemporary perspective that largely focuses on organismal diversity or abundances of gene families. Variation in diversity across component steps of key biogeochemical processes and the dynamics of within-gene diversity in response to environmental change is therefore unexplored. This hampers our ability to predict ecosystem responses to future environmental changes. To illustrate the importance of diversity across individual component steps of biogeochemical processes, we use the analogy of an electrical circuit (Fig 11). Continuity from one step to the next is required for the full process/circuit to function. To preserve integrity of the circuit there is parallelization within each component step, whereby there are multiple options for completing a given step (Condition A). In a biological context, this manifests as multiple organisms encoding the same activity through different alleles of the same gene. Under different environmental conditions, various options may not be available either because the conditions are not favorable to the expression or operation of the gene, or the organism encoding that gene is eliminated from the community. The function is maintained by the availability or

introduction of alternates that can function under the new conditions (Condition B). Conditions may exist, however, under which no options for a given component step are available to the system, for example if an anaerobic system was exposed to sufficient oxygen to inhibit nitrous oxide reductase activity. This scenario will prevent the full biogeochemical process (e.g., denitrification) from completing, at least temporarily, even if some component steps are functioning (Condition C). Steps with low within-gene diversity are more likely to experience environmental conditions that cause all options to be eliminated. Just as a chain is only as strong as its weakest link, the ability of a metabolic pathway to continue functioning is determined by the component step with the lowest diversity.

We propose that accounting for the influence of environmental variation on realized biogeochemical rates in predictive models should connect environmental conditions to the dynamics of component steps. Doing so would allow models to account for variation in the susceptibility of each step to perturbation, based on within-gene diversity and dynamics. For example, reaction network models could represent the combined influence of gene-level abundance and diversity on continued function during and after perturbation. Recent modeling developments open up such opportunities, such as Song et al.'s reaction network model that explicitly represents control of enzyme expression at each step along a given biogeochemical pathway [39]. This model could be easily modified to represent different levels of diversity and abundance of gene phylotypes across component steps. Numerical experiments using the resulting model could comprehensively explore the sensitivity of biogeochemical function to among-step variation in within-gene diversity and dynamics. We also contend that there is a need to incorporate within-gene diversity into our conceptualization of diversity and focus on understanding the ecological processes governing diversity within individual genes. Merging such ecological knowledge with mechanistic biogeochemical models should improve our ability to predict biogeochemical function under future environmental conditions.

## Experimental procedures

### Sampling

Sediment communities were captured using sand packs incubated within piezometers as described [20]. Briefly, 1.2 m, fully-screened, stainless steel piezometers (5.25 cm inner diameter) (S5a Fig) were deployed along the margin of the Columbia River at approximately 46˚ 22' 15.80"N, 119˚ 16' 31.52"W. Sand packs composed of ~80 cm$^3$ of locally-sourced medium grade sand (>0.425mm <1.7mm) packed into 2 x 4.5", 18/8 mesh stainless steel infuser plugged with Pyrex fiber glass (S5b Fig) were sterilized by combustion at 450˚C for 8hr and then deployed in pairs for six week incubations collected at three week intervals from April 30, 2014 to November 25, 2014. Upon retrieval, paired sand packs were combined and homogenized. A ~145 mL subsample was flash-frozen and transported on dry ice back to the laboratory for metagenomic analysis. Aqueous samples were taken as previously described [20]. Briefly, at each piezometer, peristaltic pumps and manifolds were purged for 10–15 minutes. Following the purge, water was pumped through 0.22 μm polyethersulfone Sterivex filters for 30 minutes. Filtered water was used for water chemistry analysis.

Sampling equipment was installed after required consultations and permits were obtained from appropriate state and federal agencies, including the Department of Energy's Pacific Northwest Site Office, the U.S. Fish and Wildlife Service, the National Marine Fisheries Service, the U.S. Army Corps of Engineers, and the Washington Department of Fish and Wildlife. All federal requirements under the National Environmental Policy Act were followed.

## Water chemistry

Water chemistry was determined as previously described [20]. Briefly, water temperature was measured with a handheld meter (Ultrameter II, Myron L Co Carlsbad, CA). A YSI Pro ODO handheld with an optical DO probe (YSI Inc. Yellow Springs, OH) was used to measure dissolved oxygen. NPOC was determined by the combustion catalytic oxidation/NDIR method using a Shimadzu TOC-Vcsh with ASI-V auto sampler (Shimadzu Scientific Instruments, Columbia, MD). Samples were acidified with 2 N HCl and sparged for 5 minutes to remove DIC. The sample was then injected into the furnace set to 680˚C. Nitrate concentrations were determined on a Dionex ICS-2000 anion chromatograph with AS40 auto sampler. A 25-minute gradient method was used with a 25-µL injection volume and a 1 mL/min flow rate at 30˚C (EPA-NERL: 300.0).

## DNA extraction

Genomic DNA was prepared from piezometer T4 sediment samples as previously described [20]. Briefly, to release biomass, thawed samples were suspended in 20mL of chilled PBS /0.1% Na-pyrophosphate solution and vortexed for 1 min. The suspended fraction was decanted to a fresh tube and centrifuged for 15' at 7000 x $g$ at 10˚C. DNA was extracted from the resulting pellets using the MoBio PowerSoil kit in plate format (MoBio Laboratories, Inc., Carlsbad, CA) following manufacturer's instructions with the addition of a 2-hour proteinase-K incubation at 55˚C prior to bead-beating to facilitate cell lysis. Subsamples of each preparation were used for 16S rRNA amplicon sequencing and shotgun metagenomic sequencing.

## Sequencing

Genomic DNA purified from sandpack samples was submitted to the Joint Genome Institute under JGI/EMSL proposal 1781 for paired-end sequencing on an Illumina HiSeq 2500 sequencer. Results from the sequencing are presented in S1 Table. Data sets are available through the JGI Genome Portal (http://genome.jgi.doe.gov). Project identifiers are listed in S1 Table.

For the 16S rRNA amplicon analysis, the protocol developed by the Earth Microbiome Project (http://press.igsb.anl.gov/earthmicrobiome/emp-standard-protocols/16s/) was followed, with the exception that the twelve base barcode sequence was included in the forward primer. Amplicons were sequenced on an Illumina MiSeq using the 300 cycle MiSeq Reagent Kit v2 (http://www.illumina.com/) according to manufacturer's instructions.

## Metagenomic analysis

To quantitate gene families of interest, hidden Markov models (HMMs) were obtained or built and searched against raw metagenomic reads. HMMs used in this study are listed in Table 2. HMMs were searched against raw reads using MaxRebo (Lee Ann McCue, unpubl.), which translates each read in six frames, and searches the translations against the target HMM(s), using HMMer [40] on a distributed, high-performance computing framework. Output was screened for reads with a significant score (e-value ≤ 1e-25) against the HMM. Raw counts were converted to RPKM (reads per kilobase of gene length per million reads) using the HMM length x 3 as the gene length. Results from forward and reverse reads were averaged and normalized against the summed RPKMs of the rplB and rplB_arch models. Individual genes of interest were assembled from the combined metagenomic datasets using the Xander assembler [41] and the HMMs listed in Table 2 and associated required files. Resulting contigs were clustered at 90% amino acid identity (Supplementary Data 1) to define phylotypes. Phylogeny was

**Table 2. HMMs used in this study.**

| Gene | | HMM | Source |
|---|---|---|---|
| **Nitrate reductase, alpha subunit (narG)** | | narG | FunGene[1] |
| **Cu-containing nitrite reductase (nirK)** | | | |
| | **Clade I** | nirK1 | PNNL[2] |
| | **Clade II** | nirK2 | PNNL |
| **Fe-containing nitrite reductase (nirS)** | | nirS | FunGene |
| **Nitric oxide reductase (norB)** | | | |
| | **Copper** | norB_cNor | FunGene |
| | **quinone** | norB_qNor | FunGene |
| **Nitrous oxide reductase** | | | |
| | **nosZI** | nosZ | FunGene |
| | **'non-denitrifying' (nosZII)** | nosZ_a2 | FunGene |
| **Ammonia monooxygenase** | | | |
| | **bacterial** | amoA_AOB | FunGene |
| | **archaeal** | amoA_AOA | FunGene |
| **Nitrite oxidoreductase, alpha subunit** | | nxrA-1 | PNNL |
| **Ribosomal protein RplB** | | | |
| | **bacterial** | rplB | FunGene |
| | **archaeal** | rplB_arch | PNNL |

[1] Available at https://github.com/rdpstaff/Xander_assembler

[2] Available at https://github.com/wichne/Xander_files

assessed by aligning protein sequences with mafft v7.164b [42, 43] and constructing approximated maximum-likelihood trees using FastTree v2.1.9 [44]. Phylotype abundance profiles were determined by searching individual metagenomic read sets against the resulting gene contigs and calculating RPKM values and normalizing against the summed phylotype RPKM for the gene. Bray-Curtis dissimilarity between samples for each gene was calculated using the R package vegan [45], and resulting values were used to generate a boxplot.

## Community analysis

Amplicon data used was from Graham et al., 2016b. Bray-Curtis distance was determined as described below, and plotted using R.

## Statistics

Bray-Curtis dissimilarity, as implemented in the R package vegan [45], was used to measure beta diversity. Values were averaged for both the total dataset and the T4 dataset alone. Early (n = 6) versus late (n = 5) gene abundance comparisons were tested for significance using the Mann-Whitney-Wilcoxon test as implemented in R v.3.3.2 (https://www.r-project.org). For turnover heatmaps, assembled sequences were searched against the read set to estimate individual abundances. Sequences were then clustered into phylotypes at 90% identity, and abundances summed. The relative abundance of each phylotype was then determined by dividing its abundance by the summed abundance of all phylotypes of the gene in question. Trees were determined from nucleic acid sequence alignments (mafft v) using the maximum-likelihood approach implemented in FastTree. Inverse Simpson statistic for the assembled sequences was calculated cumulatively for each gene at each time point, also using the vegan package. Linear regressions and associated $R^2$ and p-values were calculated in R v3.3.2.

## Supporting information

**S1 Fig. Environmental parameter correlation.** Temperature (Temp), dissolved oxygen (DO), chloride ion concentration (Cl), sulfate concentration (SO4), nitrate concentration (NO3) and dissolved organic carbon (measured as non-purgeable organic carbon, NPOC) measurements were taken for all samples. Pair-wise correlation of observations were performed to determine the independence of the parameters.
(PDF)

**S2 Fig. Environmental parameter vs diversity (as measured by the inverse Simpson statistic) linear regression analysis.** Blue borders: $p < 0.10$; Orange borders: $p < 0.05$.
(PDF)

**S3 Fig. Environmental parameter vs gene abundance linear regression analysis.** Blue borders: $p < 0.10$; Orange borders: $p < 0.05$.
(PDF)

**S4 Fig. Environmental parameter vs richness linear regression analysis.** Blue borders: $p < 0.10$; Orange borders: $p < 0.05$.
(PDF)

**S5 Fig. Sampling setup.** (a) **S**tainless steel piezometers (5.25 cm inner diameter) that were fully-screened for 1.2 m were driven into the river bottom sediment. (b) 4.5" **s**tainless steel infusers (18/8 mesh) were packed with ~80 $cm^3$ of locally-sourced medium grade sand (>0.425mm <1.7mm) and plugged with Pyrex fiber glass. Paired sand packs were deployed as shown in panel a) for six week incubations collected at three week intervals from April 30, 2014 to November 25, 2014.
(PDF)

**S1 Table. Metagenomic sequence data sets.**
(PDF)

## Acknowledgments

A portion of the research was performed using Institutional Computing at PNNL.

## Author Contributions

**Conceptualization:** William C. Nelson, Emily B. Graham, James C. Stegen.

**Data curation:** William C. Nelson.

**Formal analysis:** William C. Nelson.

**Funding acquisition:** James C. Stegen.

**Investigation:** William C. Nelson, Emily B. Graham, Alex R. Crump, Sarah J. Fansler, Evan V. Arntzen, David W. Kennedy.

**Methodology:** William C. Nelson.

**Project administration:** James C. Stegen.

**Supervision:** James C. Stegen.

**Writing – original draft:** William C. Nelson, Emily B. Graham, James C. Stegen.

**Writing – review & editing:** William C. Nelson, Emily B. Graham, James C. Stegen.

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
