## [Decision Letter · Decision Letter 0]

3 Dec 2019

PONE-D-19-29302

Distinct temporal diversity profiles for nitrogen cycling genes in a hyporheic microbiome

PLOS ONE

Dear Dr. Nelson,

Thank you for submitting your manuscript to PLOS ONE. After careful consideration, we feel that it has merit but does not fully meet PLOS ONE’s publication criteria as it currently stands. Therefore, we invite you to submit a revised version of the manuscript that addresses the points raised during the review process.

We would appreciate receiving your revised manuscript by Jan 17 2020 11:59PM. To enhance the reproducibility of your results, we recommend that if applicable you deposit your laboratory protocols in protocols.io, where a protocol can be assigned its own identifier (DOI) such that it can be cited independently in the future. For instructions see: http://journals.plos.org/plosone/s/submission-guidelines#loc-laboratory-protocols

We look forward to receiving your revised manuscript.

Kind regards,

Wenzhi Liu

Academic Editor

PLOS ONE

Journal Requirements:

1. Please include a copy of Table 2 which you refer to in your text on page 18

Reviewers' comments:

Reviewer's Responses to Questions

**Comments to the Author**

1. Is the manuscript technically sound, and do the data support the conclusions?

Reviewer #1: Yes

Reviewer #2: Yes

2. Has the statistical analysis been performed appropriately and rigorously? 

Reviewer #1: Yes

Reviewer #2: Yes

3. Have the authors made all data underlying the findings in their manuscript fully available?

Reviewer #1: Yes

Reviewer #2: Yes

4. Is the manuscript presented in an intelligible fashion and written in standard English?

Reviewer #1: Yes

Reviewer #2: Yes

5. Review Comments to the Author

Reviewer #1: The manuscript clearly illustrated the seasonal dynamic profiles of N-cycling gene diversity in a hyporheic microbiome. It is well written and the results and discussion are presented well, except for the methods which should be described in more detail. Especially for figure 10, the analogy of an electrical circuit is very imagery and easy for readers to understand the importance of diversity in biogeochemical processes. Thus, I would suggest minor revision.

Some of specific comments:

1． Line 76, please illustrate the reason(s) of designing the three locations (T2, T3 and T4) and the differences between them.

2． Line 105-118, in this paragraph, too many details on the determination of diversity are given in results, which is suggested to put into the methods.

3． Line 347, to better understand the sediment and aqueous sample collection, the sampling device schematic is suggested to be attached.

4． How many samples were assigned for DNA extraction? Which DNA was assigned for sequencing (including the fraction of 16S rRNA gene and Metagenomic analysis)? And which area of 16S rRNA gene was sequenced? It is necessary to be elaborated clearly in the methods.

5． In addition, gene names should be written in the formal way, such as, change 16S to 16S rRNA gene in figure 2, and write the gene names, such as narG in italic. The time ruler in the figures is recommended to be consistent.

Reviewer #2: Overall a well written paper with sufficient data to support claims. I recommend the paper be published in PLOS ONE, but have some minor revision suggestions for increasing clarity to future readers. Each suggestion is listed below:

1. For all of the heat map figures (3-6), the current image quality is low and it is impossible to read the individual gene phylotype names.

Also, in these figures some of the boxes are shaded gray - it is unclear what the gray shade means/represents, it would be great if this could be clarified in the figure legend.

In my opinion, I do not think the heat map figures do a good job of clearly showing changes in gene phylotype changes over time. Is it possible to graph this data another way? I was thinking an alluvial diagram would be a better way to visualize the phylotype changes. I think better graphics would increase reader accessibility.

2. As currently written, it is hard for non-experts to follow the results descriptions of nitrification and denitrification genes and keep straight which genes belong to which metabolic pathway. Adding short clarifying sentences would help.

For example, in line 120 it would be helpful to add a sentence about denitrification genes being discussed - similar to what is said about nitrification around line 128-129.

If there is space, it would be useful to create a diagram that lists the steps of nitrification and denitrification in sequential order and bold/highlight each of the genes in these metabolisms you searched for during your analyses. It would be great if this diagram was in the introduction so the reader could always refer to it. This sort of graphic would help guide readers as to why it is unexpected nxrA showed little correlation with amoA (line 175-176).

In the discussion (line 209-211, 223-225) you mention breaking the nitrification and denitrification process into component steps, yet these steps are unclear in the results, as it seems each gene is discussed individually. Adding a sentence in the results identifying which genes make up component steps and highlighting these components steps in the proposed nitrification/denitrification figure listed above would be useful for readers to follow your work.

3. Line 146-148 is confusing and unclear yet seemingly a critical conclusion from this work. You state that Figure 8 suggests that establishment of novel organisms, with the genes you looked at, is rare. How do you reach this conclusion? Would new unique microbes with the genes you are interested in cause an increase in diversity and a steeper curve in on the inverse Simpson plot? A brief justification would be useful.

4. If space, please spell out what RPKM stands for in Figure legend 9 (line 179).

5. A small typo exists at the end of line 215-216.

6. Line 405 - the HMM table needs to be table 2, and this number needs to be corrected in line 396

6. PLOS authors have the option to publish the peer review history of their article (what does this mean?). If published, this will include your full peer review and any attached files.

Reviewer #1: No

Reviewer #2: Yes: Anne E. Booker

---

## [Author Response · Author response to Decision Letter 0]

6 Jan 2020

Journal Requirements:

The manuscript has been revised and file names have been changed to meet PLOS ONE's style requirements.

1. Please include a copy of Table 2 which you refer to in your text on page 18

Table 2 was mislabeled in the initial submission. The error has been corrected.

2. Please include captions for your Supporting Information files at the end of your manuscript, and update any in-text citations to match accordingly.

The requested text has been added.

Review Comments to the Author

Reviewer #1:

The manuscript clearly illustrated the seasonal dynamic profiles of N-cycling gene diversity in a hyporheic microbiome. It is well written and the results and discussion are presented well, except for the methods which should be described in more detail. Especially for figure 10, the analogy of an electrical circuit is very imagery and easy for readers to understand the importance of diversity in biogeochemical processes. Thus, I would suggest minor revision.

Some of specific comments:

1. Line 76, please illustrate the reason(s) of designing the three locations (T2, T3 and T4) and the differences between them.

The three locations were spatially-separated replications sampling the hyporheic zone. This has been noted in the text.

2. Line 105-118, in this paragraph, too many details on the determination of diversity are given in results, which is suggested to put into the methods.

We felt that it is important for the reader to understand the data types used in the work to properly assess our analysis, thus we decided to emphasize the methods used to generate the sequences and how we define ‘phylotype’ in this work.

3. Line 347, to better understand the sediment and aqueous sample collection, the sampling device schematic is suggested to be attached.

A supplemental figure portraying the sampling setup has been added to the manuscript.

4. How many samples were assigned for DNA extraction? Which DNA was assigned for sequencing (including the fraction of 16S rRNA gene and Metagenomic analysis)? And which area of 16S rRNA gene was sequenced? It is necessary to be elaborated clearly in the methods.

T4 is the only site for which we are presenting sequence data. For each timepoint, two sandpacks were collected, combined and homogenized prior to DNA extraction. Genomic DNA preparations were divided and subjected to amplicon and shotgun sequencing. The V4 region of the 16S rRNA gene was used for amplicon analysis, as proscribed by the Earth Microbiome Project. The Methods section has been edited to include this information.

5. In addition, gene names should be written in the formal way, such as, change 16S to 16S rRNA gene in figure 2, and write the gene names, such as narG in italic. The time ruler in the figures is recommended to be consistent.

We feel it is fair to use abbreviations in the key for Figure 2 (now Figure 3), however, we have altered the legend text to directly associate the abbreviations used with the formal gene names. Gene names in the text have been italicized, and figures have been updated to have uniform time scale axes.

Reviewer #2: 

Overall a well written paper with sufficient data to support claims. I recommend the paper be published in PLOS ONE, but have some minor revision suggestions for increasing clarity to future readers. Each suggestion is listed below:

1. For all of the heat map figures (3-6), the current image quality is low and it is impossible to read the individual gene phylotype names. 

Also, in these figures some of the boxes are shaded gray - it is unclear what the gray shade means/represents, it would be great if this could be clarified in the figure legend.

Presentation of large data sets is always challenging. We have supplied figures in the highest resolution specified by the journal. Perhaps a solution is to put larger versions of the figures in Supplemental Information. We have requested guidance from the editor on this matter. 

We have stipulated in the Figure 3 (now Fig 4) legend that gray shading indicates no observation of the phylotype at that timepoint.

2. In my opinion, I do not think the heat map figures do a good job of clearly showing changes in gene phylotype changes over time. Is it possible to graph this data another way? I was thinking an alluvial diagram would be a better way to visualize the phylotype changes. I think better graphics would increase reader accessibility.

An alluvial diagram is an interesting idea, but in practice, our data is more numerical than categorical, and we feel that the number of phylotypes being tracked in the figures would strain the interpretability of such a figure.

3. As currently written, it is hard for non-experts to follow the results descriptions of nitrification and denitrification genes and keep straight which genes belong to which metabolic pathway. Adding short clarifying sentences would help. For example, in line 120 it would be helpful to add a sentence about denitrification genes being discussed - similar to what is said about nitrification around line 128-129.

If there is space, it would be useful to create a diagram that lists the steps of nitrification and denitrification in sequential order and bold/highlight each of the genes in these metabolisms you searched for during your analyses. It would be great if this diagram was in the introduction so the reader could always refer to it. This sort of graphic would help guide readers as to why it is unexpected nxrA showed little correlation with amoA (line 175-176).

We have edited the text to give more context to the functions of the genes in question. In addition, we have added a summary figure (Figure 1) delineating the denitrification and nitrification processes.

4. In the discussion (line 209-211, 223-225) you mention breaking the nitrification and denitrification process into component steps, yet these steps are unclear in the results, as it seems each gene is discussed individually. Adding a sentence in the results identifying which genes make up component steps and highlighting these components steps in the proposed nitrification/denitrification figure listed above would be useful for readers to follow your work.

The component steps are simply the individual enzymatic reactions that compose the pathway, which is why each gene is discussed individually.

5. Line 146-148 is confusing and unclear yet seemingly a critical conclusion from this work. You state that Figure 8 suggests that establishment of novel organisms, with the genes you looked at, is rare. How do you reach this conclusion? Would new unique microbes with the genes you are interested in cause an increase in diversity and a steeper curve in on the inverse Simpson plot? A brief justification would be useful.

This figure has been expanded and the text has been edited to better relate the changes in phylotype composition for each gene from both a richness and diversity (inverse Simpson) perspective.

6. If space, please spell out what RPKM stands for in Figure legend 9 (line 179).

This edit has been made.

7. A small typo exists at the end of line 215-216.

The text has been edited.

8. Line 405 - the HMM table needs to be table 2, and this number needs to be corrected in line 396

The text has been edited accordingly.

---

## [Editor Report · Decision Letter 1]

9 Jan 2020

Distinct temporal diversity profiles for nitrogen cycling genes in a hyporheic microbiome

PONE-D-19-29302R1

Dear Dr. Nelson,

We are pleased to inform you that your manuscript has been judged scientifically suitable for publication and will be formally accepted for publication once it complies with all outstanding technical requirements.

With kind regards,

Wenzhi Liu

Academic Editor

PLOS ONE
---

## [Editor Report · Acceptance letter]

10 Jan 2020

PONE-D-19-29302R1 

Distinct temporal diversity profiles for nitrogen cycling genes in a hyporheic microbiome 

Dear Dr. Nelson:

I am pleased to inform you that your manuscript has been deemed suitable for publication in PLOS ONE. Congratulations! Your manuscript is now with our production department. 

With kind regards,

on behalf of

Dr. Wenzhi Liu 

Academic Editor

PLOS ONE